# SARS-CoV-2 Detection and Culture in Different Biological Specimens from Immunocompetent and Immunosuppressed COVID-19 Patients Infected with Two Different Viral Strains

**DOI:** 10.3390/v15061270

**Published:** 2023-05-29

**Authors:** Maria Cássia Mendes-Correa, Matias Chiarastelli Salomão, Fábio Ghilardi, Tania Regina Tozetto-Mendoza, Lucy Santos Villas-Boas, Anderson Vicente de Paula, Heuder Gustavo Oliveira Paiao, Antonio Charlys da Costa, Fábio E. Leal, Andrea de Barros Coscelli Ferraz, Flavia C. S. Sales, Ingra M. Claro, Noely E. Ferreira, Geovana M. Pereira, Almir Ribeiro da Silva, Wilton Freire, Evelyn Patricia Sánchez Espinoza, Erika R. Manuli, Camila M. Romano, Jaqueline G. de Jesus, Ester C. Sabino, Steven S. Witkin

**Affiliations:** 1Departamento de Molestias Infecciosas e Parasitarias, Aculdade de Medicina, Universidade de São Paulo, Av. Dr. Enéas Carvalho de Aguiar, n. 470, São Paulo 05403-000, Brazil; matias.salomao@hc.fm.usp.br (M.C.S.);; 2Hospital das Clínicas, Faculdade de Medicina, Universidade de Sao Paulo, São Paulo 05403-010, Brazil; 3Instituto de Medicina Tropical, Faculdade de Medicina, Universidade de São Paulo, São Paulo 05403-000, Brazil; 4Rua Peixoto Gomide, 645, Sao Paulo 01409-002, Brazil; 5Faculdade de Medicina da, Universidade Municipal de Sao Caetano do Sul, São Paulo 09521-160, Brazil; 6Programa de Oncovirologia, Instituto Nacional de Câncer, Rio de Janeiro 20230-130, Brazil; 7Department of Obstetrics and Gynecology, Weill Cornell Medicine, New York, NY 10065, USA

**Keywords:** SARS-CoV-2, infectiousness, cell culture, viral shedding, viral load dynamics, persistence, COVID-19

## Abstract

Introduction—The dynamics of SARS-CoV-2 shedding and replication in humans remain incompletely understood. Methods—We analyzed SARS-CoV-2 shedding from multiple sites in individuals with an acute COVID-19 infection by weekly sampling for five weeks in 98 immunocompetent and 25 immunosuppressed individuals. Samples and culture supernatants were tested via RT-PCR for SARS-CoV-2 to determine viral clearance rates and in vitro replication. Results—A total of 2447 clinical specimens were evaluated, including 557 nasopharyngeal swabs, 527 saliva samples, 464 urine specimens, 437 anal swabs and 462 blood samples. The SARS-CoV-2 genome sequences at each site were classified as belonging to the B.1.128 (ancestral strain) or Gamma lineage. SARS-CoV-2 detection was highest in nasopharyngeal swabs regardless of the virus strain involved or the immune status of infected individuals. The duration of viral shedding varied between clinical specimens and individual patients. Prolonged shedding of potentially infectious virus varied from 10 days up to 191 days, and primarily occurred in immunosuppressed individuals. Virus was isolated in culture from 18 nasal swab or saliva samples collected 10 or more days after onset of disease. Conclusions—Our findings indicate that persistent SARS-CoV-2 shedding may occur in both competent or immunosuppressed individuals, at multiple clinical sites and in a minority of subjects is capable of in vitro replication.

## 1. Introduction

Although much progress has been made in the field of SARS-CoV-2 diagnostics, to date the dynamics of SARS-CoV-2 replication, shedding and infectivity in humans remain incompletely understood [1]. Infectivity by any viral pathogen is a very complex process that involves multiple host and viral factors. Different studies on SARS-CoV-2 infection have suggested that the magnitude and duration of viral shedding correlates with biological characteristics of the virus, disease severity, patient age and sex, viral load, stage of the infection and immune status of the infected individual [1,2,3,4,5]. 

SARS-CoV-2 RNA has been detected in different body fluids in addition to nasal swabs during acute infection, including saliva, peripheral blood, ocular secretions, anal swabs, and urine [1,2,3,4,5,6,7,8,9,10,11,12,13,14,15]. A recent meta-analysis found that the mean shedding time of SARS-CoV-2 in the upper respiratory tract, lower respiratory tract, stools, and serum was 17.0, 14.6, 17.2, and 16.6 days, respectively [5]. However, results from multiple investigations have been very heterogeneous [8]. Most of these studies used only a single time point to collect samples, included limited sites of biologic specimen collection, and only utilized patients with severe forms of COVID-19 [5,9,10]. As a result, variations in time between symptom onset and sample testing can be a confounding factor when analyzing data from these studies on viral shedding and infectivity. In addition, virus isolation from non-respiratory tract specimens has been unsuccessful in most cases [1,9,10,11,12,13,14,15].

To investigate SARS-CoV-2 persistence and replication competence in individuals with mild or severe symptoms, we carried out the present study on 125 COVID-19 cases. The dynamics of infectious virus shedding from multiple biologic sites during acute infection were determined through weekly longitudinal sampling in 123 individuals. We estimated viral clearance rates and potential infectivity at multiple sites, among immunocompetent and immunosuppressed individuals with mild or severe COVID-19 disease. In addition, the shedding dynamics of the B.1.1.28 lineage (ancestral strain) and the Gamma strains of SARS-CoV-2 were compared.

The objective of the study was to investigate the presence and potential infectivity of SARS-CoV-2 in nasopharyngeal swabs, saliva, urine, blood, and feces (anal swab), among immunocompetent and immunocompromised patients, during the acute and convalescence phases of COVID-19, in patients infected with two different virus strains.

## 2. Materials and Methods

### 2.1. Study Design and Population

This was an observational prospective study, developed in patients with SARS-CoV-2 infection, using a convenience sample. Patients were followed-up for 5 weeks after their initial COVID-19 diagnosis, or until SARS-CoV-2 testing was negative in the collected samples.

Two groups of patients were included: (1) Immunocompetent patients with mild disease who were infected with either the ancestral SARS-CoV-2 (B.1.1.28 lineage) or the Gamma variant; and (2) Immunocompromised patients, hospitalized with severe disease infected with either the ancestral SARS-CoV-2 (B.1.1.28 lineage) or the Gamma variant. Hospitalized patients were seen either at the Hospital das Clínicas da Faculdade de Medicina da Universidade de São Paulo (USP) (a public hospital) or at Hospital 9 de Julho (a private hospital) in São Paulo, Brazil, between May 2020 and March 2021. All immunocompromised patients included in the study were individuals with a previous diagnosis of different types of neoplasms and were under immunosuppressive therapy at the time they were infected with SARS-CoV-2. The outpatient subjects were seen between March and May 2020. They were participants in *The Corona São Caetano Program*, a primary care initiative offering COVID-19 care to all residents of São Caetano do Sul, Brazil [16]. None of the participants received any COVID-19-specific anti-viral therapy or vaccines for SARS-CoV-2, before inclusion in the study. Age, sex, disease severity, clinical variables, and date of the onset of symptoms were retrieved from medical records stored in the laboratory information system of the two hospitals and from *The Corona São Caetano Program.*

### 2.2. Laboratory Tests and Sample Collection

#### 2.2.1. Sample Collection

All recruited patients, aged >18 years, had a confirmed diagnosis of COVID-19, through the identification of SARS-CoV-2 by RT-PCR from a nasopharyngeal swab.

After their initial COVID-19 diagnosis, during a 5-week period, with a 7-day interval, each included patient was invited to collect different samples as follows: anal and nasal swab, saliva, urine, and blood. All samples were then subjected to SARS-CoV-2 testing by RT-PCR.

#### 2.2.2. Virus Identification: RNA Extraction, PCR Amplification

All specimens, handled according to laboratory biosafety guidelines, were subjected to total nucleic acid extraction with the QIAamp viral RNA kit (QIAGEN, Hilden, Germany), according to the manufacturer’s instructions. Samples were then subjected to RT-PCR.

#### 2.2.3. Real-Time PCR for SARS-CoV-2

Quantitative assays (SARS-CoV-2 N or E gene) for SARS-CoV-2 were performed according to protocols adapted with primers and probes for the RT-PCR assay [17,18,19]. All samples were deemed suitable for amplification by RT-PCR based on the analysis of the internal control consisting of primers and probe for the human Ribonuclease P gene [20]. Sequences of oligonucleotides were resuspended in known concentrations (serial dilution to base 10) for use as a positive control and for the construction of viral load quantification curves. The synthetic oligo sequences designed for the current study were based on a protocol previously described for other pathogens [21]. The sequences of specific oligos are described in Table 1.

#### 2.2.4. Viral Culture

Viral culture for SARS-CoV-2, conducted in a biosafety level-3 facility, utilized Vero CCL81 cells (ATCC^®^ CCL-81™) in Dulbecco minimal essential medium supplemented with 5% heat-inactivated fetal bovine serum and antibiotics/antimycotics. SARS-CoV-2 PCR-positive samples were inoculated into a Vero cell culture in plastic bottles (Jet biofilm, 12.5 cm^2^ area, 25 mL capacity) and incubated in a 37 °C incubator in an atmosphere of 5% CO_2_. Cultures were maintained for at least 2 weeks and observed daily for evidence of cytopathic effects (CPEs). At least two subcultures were performed on each sample. The detection of CPEs was investigated using an inverted microscope (Nikon, Nikon, Japan) and the presence of virus in supernatants from cultures showing CPEs was determined by specific RT-PCR, as described above. Viral isolation (culture) was performed on all RT-PCR positive samples 10 or more days after onset of symptoms. The Cts of supernatant and the original clinical sample (Ct sample) were compared, and positive cultures were defined where Ct sample—Ct culture was ≥3. Culture positivity was utilized as a proxy for infectivity.

#### 2.2.5. SARS-CoV-2 Whole-Genome Sequencing

The viral RNA, extracted as described above, was also used for whole-genome sequencing (WGS) analysis. In brief, SARS-CoV-2 complementary DNA and multiplex PCR steps were performed, and the amplicons were sequenced using the MinION platform (Oxford Nanopore Technologies, UK) and Miseq (Illumina, San Diego, CA, USA) for lineage characterization [22]. Variant calling and consensus sequences were performed using artic minion with Nanopolish version from the ARTIC bioinformatics pipeline (https://github.com/artic-network/fieldbioinformatics, accessed on 1 July 2022). Genome regions with a depth of 50 times genome coverage were used for lineage classification via Pangolin version 3.1.5 (http://pangolin.cog-uk.io/, accessed on 1 July 2022) [23] and Nextclade version 1.4.0 (https://clades.nextstrain.org, accessed on 1 July 2022) and confirmed using manual genotyping.

#### 2.2.6. Co-Infection with Influenza A and B Virus and Syncytial Respiratory Virus (SRV)

All individuals from Group 1 and Group 2 were tested (nasal swab samples collected at week 1) for Influenza A and B virus and syncytial respiratory virus (SRV) at our laboratory, using the Allplex™ SARS-CoV-2/FluA/FluB/RSV Assay.

### 2.3. Definitions

The persistence of SARS-CoV-2 was defined based on the time spam (days) from the onset of symptoms to the last positive results that samples remained RT-PCR positive. Individuals with RT-PCR positive samples in any biologic specimen 10 or more days after onset of symptoms were deemed to have a persistent infection. The ability to propagate virus in an in vitro culture was used as a proxy for infectivity, as proposed by Wolfel et al. [9].

### 2.4. Statistics 

Differences between immunocompetent and immunosuppressed subjects, infection with the B1 or Gamma SARS-CoV-2 strains and viral detection between biological sites were analyzed using Fisher’s exact test or the Mann–Whitney test, as appropriate. The quantitative parameters of viral detection were described for each infecting SARS-CoV-2 strain and subjects’ immune status by week of collection using absolute and relative frequencies. Persistence of viral detection for more than 10 days in any sample was described according to clinical and demographic characteristics and the association of persistence with qualitative characteristics was verified using the chi-square test or Fisher’s exact test. Quantitative characteristics were compared according to persistence using the student t test or Mann–Whitney test. The analyses were performed using the IBM-SPSS for Windows version 20.0 software and tabulated using the Microsoft-Excel 2003 software. All tests were performed with a significance level of 5%.

## 3. Results

Ninety-eight immunocompetent patients and twenty-five immunosuppressed individuals were included in the study. Characteristics of the study population is shown in Table 2. The immunocompetent group were younger (42.7 vs. 57.4 years old, *p* < 0.0001), had a higher mean body mass index (27.4 vs. 24.9 kg/m^2^, *p* = 0.0402) and a higher mean log_10_ SARS-CoV-2 viral load (5.6 vs. 4.0, *p* = 0.0014) than the immunosuppressed group. The viral genome sequences obtained from all clinical specimens from immunocompetent patients with mild forms of disease belonged to either the B.1.128 lineage (51 cases) or the Gamma lineage (47 cases). The Gamma strain emerged in Brazil in 2020 [24,25,26]. Similarly, among immunosuppressed individuals the viral genome sequences also belonged to either the B.1.128 lineage (14 patients) or to the Gamma variant (11 patients).

None of the included patients were positive for Influenza A and B virus and syncytial respiratory virus (SRV), when we tested nasal swab samples collected at week 1.

SARS-CoV-2 B1 strain detection at different sites in immunocompetent and immunosuppressed patients is show in Table 3. The highest percentage of virus detection was in nasal samples, 35.8% of samples from immunocompetent subjects and 45.7% from immunosuppressed individuals. Saliva was the second most frequently positive site, 6.0% in immunocompetent subjects and 25.7% in the immunocompromised group. This difference was highly significant (*p* = 0.0008). Similarly, urine (*p* = 0.0002) and blood (*p* = 0.0001) samples from immunosuppressed patients were more frequently positive than from immunocompetent individuals. A small and comparable percentage of anal swabs from both groups were also positive.

SARS-CoV-2 Gamma strain detection at different sites in immunocompetent and immunosuppressed patients is detailed in Table 4. Similar to the results with the B1 strain, the highest percentage of positive samples were from the nasal cavity in both immunocompetent (42.4%) and immunosuppressed (62.5%) subjects. This difference was significant (*p* = 0.0248). Virus was also detected in a higher percentage of samples from saliva, (*p* = 0.0001), urine (*p* = 0.0001) and blood (*p* = 0.0075) from immunosuppressed than from immunocompetent individuals. A small and comparable percentage of anal swabs from both groups were also positive.

The ability to propagate the B1 strain of SARS-CoV-2 in culture from different sites in immunocompetent and immunosuppressed patients is shown in Table 5. In samples from immunocompetent individuals, virus could be cultured from only 3 of 76 nasal samples (3.9%) and 1 of 9 saliva samples (11.1%). In marked contrast, virus was cultured from 5 of 13 nasal samples (38.5%) and 3 of 8 saliva samples (37.5%) from the immunosuppressed group. This difference in virus cultivation from nasal samples was significant (*p* = 0.0014).

Among the 120 samples submitted to virus culture (Table 5), we observed contamination in 8 samples (1 anal swab and 7 nasal swabs). The contaminated isolates were submitted to two treatments. Initially, they were subjected to microfiltration using a 0.22 µ millipore filter (Minisart^®^ 17761-ACK—Sartorius Stedim) which eliminates most bacteria [27]. After this initial step, 1% penicillin-streptomycin is added to the filtered material to eliminate bacterial cells [28,29].

Among the 8 cases, after the use of this specific protocol, viral isolation was finally observed in 2 samples (nasal swab samples).

SARS-CoV-2 Gamma strain propagation in culture from samples obtained from different sites is described in Table 6. Virus was cultured from samples obtained from the nasal cavity of immunosuppressed (6 of 20, 30.0%) but not from immunocompetent (0 of 84) individuals. No virus was obtained from the cultures of saliva, urine, anal and blood from all patients.

The duration of viral shedding at different sites in immunocompetent and immunosuppressed individuals over time with prolonged viral shedding of at least 10 days is shown in Table 7. Among the immunocompetent patients, viral shedding from different sites varied from 16 days to 50 days and was longest (50 days) in nasal specimens. For immunosuppressed individuals, viral shedding varied from 177 days to 212 days and was also longest for nasal specimens (212 days).

The detection of the B1 and Gamma SARS-C0V-2 strains at different body sites over time is shown in Table 8. The highest number of positive samples was observed in nasal samples collected at week 1. The percent positive was similar for B1 (73.5%) and Gamma (76.1%) isolates. The second highest percent positive was in week 1 saliva for B1 (21.1%) and in week 1 anal swabs for Gamma (12.8%). The percentage of virus positive in saliva was significantly higher for individuals infected with B1 (21.1%) than in those infected with the Gamma strain (4.5%) (*p* = 0.0390). The B1 strain was only detected in 1/51 blood samples collected at week 1 and in none of the samples collected at weeks 2 and 3. In contrast, the Gamma strain was detected in 11.6%, 11.1% and 9.1% of blood samples collected at week 1, 2 and 3, respectively. B1 and Gamma detection in nasal samples (*p* < 0.0001), and B1 detection in saliva (*p* = 0.0148), progressively decreased over the 5-week testing period.

Analysis of all viral genome sequences obtained from clinical specimens of all patients did not reveal any mutations which would be associated with prolonged viral shedding, viral replication or pathogenicity.

## 4. Discussion

Among the immunocompetent and immunosuppressed individuals with COVID, a greater number were positive for SARS-CoV-2 in nasopharyngeal swabs than in any other sites tested regardless of the infecting viral strain or the time the sample was collected after infection initiation. At sites other than the nasopharynx—saliva, urine and blood—virus was detected more frequently in immunosuppressed individuals than in those who were immunocompetent. The few positive anal samples were not significantly different between the immunocompetent and immunosuppressed populations. Virus detection from all sites declined over time but persisted longer in immunosuppressed individuals. Viral propagation in culture was also achieved more frequently in samples from the immunosuppressed group. All these findings were consistent with earlier reports [1,30,31,32]. In immunocompetent patients, considering only samples collected ten or more days after onset of disease, 3.6%, 1.9% and 0.25% of blood, anal and urine samples, respectively, were positive for SARS-CoV-2. In contrast, among immunosuppressed individuals, considering only samples collected ten or more days after onset of disease, SARS-CoV-2 was identified in 41.17%, 5.4% and 22.9% of blood, urine and anal samples, respectively. Our data also confirmed that in immunocompetent individuals with mild COVID-19, successful virus cultivation from clinical samples obtained ten days after the onset of symptoms is an uncommon event, regardless of the viral strain involved [1,9,10,33] and that immunocompromised patients with COVID-19 are at elevated risk for prolonged viral shedding and persistent replicating capacity [33,34,35]. In our study, despite the frequent finding of SARS-CoV-2 in different biological materials and in some cases for very prolonged periods, viral replication in vitro was identified only in nasopharyngeal and saliva samples. SARS-CoV-2 has been described in different body fluids, by many authors from different parts of the world. Most of these reports, however, included only a small number of prospectively collected samples from only the nasopharynx and saliva. The present study reports findings from multiple sites in a large series of immunocompromised and immunocompetent individuals.

One of our patients, a 40-year-old male who had undergone a prior autologous hematopoietic stem cell transplant due to a diffused large B-cell lymphoma, was found to persistently shed SARS-CoV-2 that could be propagated in culture from nasal swab and saliva samples for more than 196 days. Other studies have identified atypical cases with prolonged shedding of infectious virus for up to 200 days [33,34,35,36,37].

Several prior studies have proposed that the persistence of SARS-CoV-2 capable of replication was related to immunosuppression [38,39,40]. As extensively described for numerous viral infections, both innate immunity and T cell-mediated adaptive immune response are essential for the clearance and long-term inhibition of viral infections [38,39]. It is also important to acknowledge that studies have associated the persistence of SARS-CoV-2 with the severity of disease, the presence of co-morbidities and use of glucocorticoids [41,42,43,44]. In our series of immunosuppressed patients, all were hospitalized with severe forms of COVID-19.

Viral evolution of SARS-CoV-2 over time has led to the emergence of numerous variants. Differences in immune evasion, viral loads, and duration of shedding between variants have been described [1,5]. It was not surprising, therefore, that in our study differences in viral detection, persistence, and proliferation in culture between the B1 and Gamma strains were observed.

Only a few studies have described temporal changes in SARS-CoV-2 detection [45,46,47]. The virus detection rate in blood in different studies was between 28% and 32% for hospitalized patients. However, patients in intensive care had rates up to 78% [3,45,46,47]. Our study is unique in that it describes the prospective presence of SARS-CoV-2 in non-hospitalized individuals with only mild forms of COVID-19. In addition, to our knowledge no prior investigation has reported in vitro viral propagation from SARS-CoV-2-positive blood samples.

Review studies have found overall rates of SARS-CoV-2-positive urine samples to be low and levels correlated with severe disease status [48]. The present study is in agreement with these prior reports. SARS-CoV-2 positive urine samples were identified in 0.25% of immunocompetent COVID-19 patients with mild disease, but in 22.9% of immunosuppressed patients with severe disease. None of the virus-positive samples yielded virus upon cultivation. Similarly, SARS-CoV-2 has been identified in anal swabs [6,10,11,12]. The dynamics of SARS-CoV-2 shedding at this site has been described as erratic, with the highest viral loads reported during the first weeks after symptom onset and described in severe cases [5,6,10,11,12,41]. In our study, SARS-CoV-2 detection in anal swabs was observed in 10 samples (1.9%) from immunocompetent individuals and in 5.4% of swabs from immunosuppressed individuals. Replicative virus was not obtained from any anal sample. This was in accord with findings from prior studies [5,41,49].

Much progress has been made in understanding the transmission dynamics of SARS-CoV-2 and duration of infectivity. The World Health Organization (WHO) and the Center for Disease Control (CDC) have modified their recommendations in response to data indicating that infectivity decreases to essentially zero after about 10 days from symptom onset in mild to moderately ill patients and after about 15 days in critically ill and immunocompromised patients, with a maximum reported interval thus far of 20 days [50]. To our knowledge, the present study is unique in comparing both SARS-CoV-2 detection by RT-PCR and by the presence of culturable virus in serially collected samples from different clinical sites, in both immunocompetent and immunosuppressed individuals. We also directly compared the shedding dynamics of two different SARS-CoV-2 strains—B and Gamma—in these individuals. This enabled us to estimate the time period of virus detection and its relationship to capacity for replication in different biological specimens.

It is necessary to acknowledge the limitations of our study. All of our cases occurred during the initial two years of the pandemic, before the widespread circulation of additional viral variants. Therefore, our findings might not be generalizable to current and future SARS-CoV-2 variants. Additionally, with the continuous circulation of SARS-CoV-2 in the community, previous or recurrent infection, vaccination, or a combination of both could alter viral shedding patterns differently from those observed in our study. In addition, while study staff were trained in sample collections it is possible that sample variation in quality could have occurred. However, the consistency in findings from longitudinal sampling suggest that this variation was likely minimal.

## 5. Conclusions

The present study describes the longitudinal dynamics of SARS-CoV-2 infection in immunocompetent individuals with mild disease as well as in immunocompromised individuals with severe disease. Delineation of the duration of virus detection and propagation capability in diverse biological specimens from these two different populations is fundamental to an improved understanding of contagion and development of more effective and evidence-based intervention policies. Additional studies are needed to further clarify the risk factors and features associated with persistent shedding of potentially infectious SARS-CoV-2 among other groups of individuals infected with various viral strains.

## Figures and Tables

**Table 1 viruses-15-01270-t001:** Description of the oligo sequences for the detection of CoV-2 by RT-qPCR.

Oligo	Description	Sequence 5′-3′
Forward	HKU-NF	5′-TAA TCA GAC AAG GAA CTG ATT A-3′
Reverse	HKU-NR	5′-CGA AGG TGT GAC TTC CAT G-3′
Probe	HKU-NP	format 5′-Cy5/TAO/3′-IABkRQ): 5′-GC AAA TTG TGC AAT TTG CGG-3′
Curve	Synthetic	5′-ttcgtCGAAGGTGTGACTTCCATGcgtatCCGCAAATTGCACAATTTGC atgcgtAATCAGTTCCTTGTCTGATTActgata-3′
Forward	E-Sarberco F1	5′- ACA GGT ACG TTA ATA GTT AAT AGC GT-3′
Reverse	E-sarberco-R2	5′-ATA TTG CAG CAG TAC GCA CAC A-3′
Probe	E_sarberco P1	format 5′-VIC/ZEN/3′IABkFQ: 5′- ACA CTA GCC ATC CTT ACT GCG CTT CG-3′
Curve	Synthetic	5′-ttcgtATATTGCAGCAGTACGCACACcgtatCGAAGCGCAGTAAGGA TGGCTAGTGTatgcgtACGCTATTAACTATTAACGTACCTGTctgata-3′

The RT PCR data were expressed as the value of the Cycle threshold (Ct), corresponding to the initial amplification cycle, which is inversely proportional to the number of copies of the target sequence of interest, given by the measurement of the number of copies per reaction.

**Table 2 viruses-15-01270-t002:** Clinical characteristics of study population.

Characteristic	Immunocompetent	Immunosuppressed	*p* Value
	N = 98	N = 25	
Mean age (SD)	42.7 (14.5)	57.4 (14.0)	<0.0001
Gender			
% male	33.7%	56.0%	0.0636
% female	66.3%	44.0%	
Mean body mass index (SD)	27.4 (5.5)	24.9 (5.4)	0.0402
Infecting SARS-CoV-2 strain (%)			
B1	51 (52.0%)	14 (56.0%)	
Gamma	47 (48.0%)	11 (44.0%)	
Mean log_10_ viral load (SD)	5.6 (1.3)	4.0 (1.9)	0.0014

**Table 3 viruses-15-01270-t003:** SARS-CoV-2 B1 strain detection at different sites in immunocompetent and immunosuppressed patients.

Site	Immunocompetent		Immunosuppressed		
	No. Submitted to	No. B1	No. Submitted to	No. B1	
	RNA Extraction	Positive (%)	RNA Extraction	Positive (%)	*p* Value
Nasal	246	88 (35.8%)	35	16 (45.7%)	0.2664
Saliva	235	14 (6.0%)	35	9 (25.7%)	0.0008
Urine	244	0	35	4 (11.7%)	0.0002
Anal	219	2 (0.9%)	35	2 (5.7%)	0.0928
Blood	244	1 (0.4%)	34	6 (17.6%)	0.0001
Total *	1188	105 (8.8%)	174	37 (21.3%)	0.0001

SARS-CoV-2 detection was by RT-PCR; * Number of samples collected at multiple time points from all included patients during the study.

**Table 4 viruses-15-01270-t004:** SARS-CoV-2 Gamma strain detection at different sites in immunocompetent and immunosuppressed patients.

Site	Immunocompetent		Immunosuppressed		
	No. Submitted to	No. Gamma	No. Submitted to	No. Gamma	
	RNA Extraction	Positive (%)	RNA Extraction	Positive (%)	*p* Value
Nasal	236	100 (42.4%)	40	25 (62.5%)	0.0248
Saliva	224	6 (2.7%)	33	20 (60.6%)	0.0001
Urine	146	1 (0.7%)	39	16 (41.0%)	0.0001
Anal	144	8 (5.6%)	39	2 (5.1%)	1.000
Blood	144	14 (9.9%)	40	11 (27.5%)	0.0075
Total *	894	129 (14.6%)	191	74 (38.7%)	0.0001

SARS-CoV-2 detection was by RT-PCR; * Number of samples collected at multiple time points from all included patients during the study.

**Table 5 viruses-15-01270-t005:** SARS-CoV-2 B1 strain propagation in culture from samples obtained from different sites.

Site	Immunocompetent		Immunosuppressed		
	No. Submitted to	No. B1	No. Submitted to	No. B1	
	Viral Culture	Positive (%)	Viral Culture	Positive (%)	*p* Value
Nasal	76	3 (3.9%)	13	5 (38.5%)	0.0014
Saliva	9	1 (11.1%)	8	3 (37.5%)	0.2941
Urine	0	0	4	0	
Anal	2	0	2	0	
Blood	1	0	5	0	
Total	88	4 (4.5%)	32	8 (25.0%)	0.0026

SARS-CoV-2 detection was by RT-PCR.

**Table 6 viruses-15-01270-t006:** SARS-CoV-2 Gamma strain propagation in culture from samples obtained from different sites.

Site	Immunocompetent		Immunosuppressed		
	No. Submitted to	No. Gamma	No. Submitted to	No. Gamma	
	Viral Culture	Positive (%)	Viral Culture	Positive (%)	*p* Value
Nasal	84	0	20	6 (30.0%)	0.0001
Saliva	5	0	16	0	
Urine	1	0	13	0	
Anal	5	0	2	0	
Blood	13	0	9	0	
Total	108	0	60	6 (10.0%)	0.0018

SARS-CoV-2 detection was by RT-PCR.

**Table 7 viruses-15-01270-t007:** Duration of SARS-CoV-2 detection at different sites in immunocompetent and immunosuppressed individuals.

Site	Days Virus Positive	
	Immunocompetent	Immunosuppressed
Nasal	10-50	10-212
Saliva	10-40	10-191
Urine	10-16	10-191
Anal	10-21	10-177
Blood	10-38	10-177

Virus detection was by RT-PCR.

**Table 8 viruses-15-01270-t008:** Detection of SARS-CoV-2 B1 and Gamma strains over time from different sites.

Virus	Site			No. Positive/No. Tested (%)		
		Week 1	Week 2	Week 3	Week 4	Week 5
B1	nasal	36/49 (73.5)	20/48 (41.7)	16/50 (32.0)	9/48 (18.8)	7/43 (16.3) ^b^
Gamma		35/46 (76.1)	26/47 (55.3)	17/45 (37.8)	9/38 (23.7)	12/37 (32.4) ^b^
B1	saliva	8/38 (21.1) ^a^	2/43 (4.7)	0/49	3/45 (6.7)	1/39 (2.6) ^c^
Gamma		2/44 (4.5)	1/41 (2.4)	1/35 (2.9)	1/41 (2.4)	1/32 (3.1)
B1	urine	0/45	0/51	0/48	NT	NT
Gamma		0/43	1/46 (2.2)	0/41	NT	NT
B1	anal	1/51 (2.0)	1/34 (2.9)	0/37	NT	NT
Gamma		5/39 (12.8)	2/42 (4.8)	0/40 ^d^	NT	NT
B1	blood	1/51 (2.0)	0/17	0/1	NT	NT
Gamma		5/43 (11.6)	5/45 (11.1)	4/44 (9.1)	NT	NT

^a^ *p* = 0.0390 vs. Gamma; ^b^
*p* < 0.0001 vs. week 1; ^c^
*p* = 0.0148 vs. week 1; ^d^
*p* = 0.0255 vs. week 1 NT, not tested.

## Data Availability

Not applicable.

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
