# Peer review of "SARS-CoV-2 Detection and Culture in Different Biological Specimens from Immunocompetent and Immunosuppressed COVID-19 Patients Infected with Two Different Viral Strains"

_viruses, 2023, doi:10.3390/v15061270_

Round 1
Reviewer 1 Report
I congratulate the authors, the work is well structured and scientifically valid. I find it very interesting since it analyzes the different ability of different biological samples to be or not infectious, as regards the SARS-CoV-2 virus. I have only a few minor considerations.
Minor questions:
1) In lines 79-81, group 2-immunocompromised patients should be better defined, specifying whether they were immunocompromised because they had pathologies such as primary immunodeficiencies or immunodeficiencies secondary to other pathologies (eg neoplasms) or because they were subjected to immunosuppressive therapies.
2) In lines 236-258 the authors show that a higher percentage of samples from immunocompromised patients allowed to isolate the virus in culture (about 37 - 38% for the B1 strain and about 30% for the gamma strain), compared to immunocompetent patients. The authors assessed whether this was related to higher viral loads in samples from immunocompromised patients? (i.e. lower ct). The authors should evaluate whether this correlation existed or not and comment on this result and try to explain it.
3) In lines 291-301 the authors highlight that the samples with the highest number of positives were the nasal samples collected at 1 week, followed by the saliva samples. So here it appears that these types of samples have a higher viral load ? Has a correlation with the ct value been evaluated ? If there is this data it should be added.
Author Response
Answers to Reviewer 1
Minor questions:
- In lines 79-81, group 2-immunocompromised patients should be better defined, specifying whether they were immunocompromised because they had pathologies such as primary immunodeficiencies or immunodeficiencies secondary to other pathologies (eg neoplasms) or because they were subjected to immunosuppressive therapies.
Answer- We thank the Reviewer for this important observation. We have modified the text to address this important point and to include the requested information in the manuscript.
All immunocompromised patients were individuals with a previous diagnosis of different types of neoplasms and were under immunosuppressive therapy at the time they were infected by SARS CoV-2.
Please check the Material and Methods section which has been rewritten (lines 87-90).
- In lines 236-258 the authors show that a higher percentage of samples from immunocompromised patients allowed to isolate the virus in culture (about 37 - 38% for the B1 strain and about 30% for the gamma strain), compared to immunocompetent patients. The authors assessed whether this was related to higher viral loads in samples from immunocompromised patients? (i.e., lower ct). The authors should evaluate whether this correlation existed or not and comment on this result and try to explain it.
Answer: We thank the Reviewer for raising this point.
In fact, when we compared the SARS CoV-2 viral load in nasal swabs from immunocompetent versus immunosuppressed individuals at week 1 of follow up, we observed that the immunocompetent group had a higher mean log10 SARS-CoV-2 viral load (5.6 vs. 4.0, p = 0.0014) than the immunosuppressed group (Table 2).
However, we did not observe any significant difference in the SARS CoV-2 viral load or Ct values when we compared nasal swab samples from both groups of patients at other time points.
Also, we did not find any association between the SARS CoV-2 viral load and prolonged viral shedding or in vitro replication capacity in our study.
On the other hand, as discussed in the text, persistent SARS-CoV-2 shedding occurred more frequently among immunosuppressed individuals.
It is plausible to suppose, as mentioned in Discussion (lines 393-400) that, in the group of patients included in our study, their immunological status was the most determinant factor for SARS CoV-2 persistence (prolonged shedding) and propagation capability in vitro. Other patient characteristics also observed among immunosuppressed individuals included in our study, such as severity of disease, the presence of co-morbidities and use of glucocorticoids [38-41] may have contributed to prolonged shedding and in vitro replication capacity. Please check Discussion (lines 393-400) for these observations.
- In lines 291-301 the authors highlight that the samples with the highest number of positives were the nasal samples collected at 1 week, followed by the saliva samples. So here it appears that these types of samples have a higher viral load? Has a correlation with the Ct value been evaluated? If there is this data, it should be added.
Answer- Indeed, as mentioned in Results the highest number of positive samples were from nasal samples collected at 1 week, followed by the saliva samples.
As a matter of fact, among immunocompetent and immunosuppressed individuals with COVID, a greater number were positive for SARS-CoV-2 in nasopharyngeal swabs than in any other sites tested regardless of the infecting viral strain or the time after infection initiation that the sample was collected (Discussion, lines 361-364).
Unfortunately, we did not compare Ct values with viral load levels in nasal swab, saliva and other biologic samples, at any time point.

Reviewer 2 Report
The present study by Mendes-Correa and colleagues investigated the shedding of SARS-CoV-2 from multiple sample sites in immunocompetent and -suppressed patients that displayed mild to severe symptoms of COVID-19. There are already several studies available focussing on viral loads and shedding of SARS-CoV-2. However, most of these studies analysed only limited sites of biological specimens that were obtained at one particular time point of infection from patients with severe symptoms of illness. Mendes-Correa et al. provide novel information on the shedding of SARS-CoV-2 in different specimens during a time span of at least five weeks.
The following points should be addressed to increase the quality of this study:
1. Line 87-88: Is there any information regarding co-infection of individual patients?
2. Line 169 – 171: “The immunocompetent group … had a higher mean log10 SARS-CoV-2 viral load (5.6 vs. 4.0, p = 0.0014) than the immunosuppressed group.” Is it true for all time points? Do you have any explanation why the immunosuppressed patients display lower viral loads but prolonged shedding?
3. Table 3 and 4: It is unclear which samples have been included in the total number of samples submitted to RNA extraction. Are those samples derived from one time point per patient or from multiple time points of sample collection? Please add this information to the table legend.
4. Did you detect any mutations in the immunosuppressed patients that showed prolonged viral shedding?
Quality of English language is fine. There are only some typing errors that have to be corrected e.g. missing or surplus/double space such as line 98 "SARS- CoV-2"
Author Response
Answers to Reviewer 2
- Line 87-88: Is there any information regarding co-infection of individual patients?
Answer- We thank the Reviewer for this observation. In fact, all individuals from Group 1 and Group 2 were tested for Influenza A and B virus and syncytial respiratory virus (SRV) at our laboratory, by the Allplex™ SARS-CoV-2/FluA/FluB/RSV Assay. None of them were positive for these viruses, when we tested nasal swab samples collected at week 1.
We have added this information to the text as requested. Please refer to Methods (lines 156-159) and Results (lines 209-210)
Hospitalized patients were eventually tested for other viruses or bacteria during their period of hospitalization, on an individual basis. We were unable to obtain the complete information on these secondary analyses.
- Line 169 – 171: “The immunocompetent group … had a higher mean log10 SARS-CoV-2 viral load (5.6 vs. 4.0, p = 0.0014) than the immunosuppressed group.” Is it true for all time points? Do you have any explanation why the immunosuppressed patients display lower viral loads but prolonged shedding?
Answer: We thank the Reviewer for raising this point. In fact, when we compared SARS CoV-2 viral load in nasal swabs from immunocompetent versus immunosuppressed individuals at week 1 of follow up, we observed that the immunocompetent group had a higher mean log10 SARS-CoV-2 viral load (5.6 vs. 4.0, p = 0.0014) than the immunosuppressed group.
However, we did not observe any significant difference in the SARS CoV-2 viral load or Ct values when we compared nasal swab samples from both groups of patients at other time points.
It is plausible to suppose, as mentioned in Discussion (lines 393-400) that, in the group of patients included in our study, the immunological status of patients was the primary determining factor for SARS CoV-2 persistence (prolonged shedding) and propagation capability in vitro. Other patient characteristics also observed among immunosuppressed individuals included in our study, such as severity of disease, the presence of co-morbidities and use of glucocorticoids [38-41] may have contributed to prolonged shedding and in vitro replication capacity. Please check Discussion (lines 393-400) for these observations.
- 3.Table 3 and 4: It is unclear which samples have been included in the total number of samples submitted to RNA extraction. Are those samples derived from one time point per patient or from multiple time points of sample collection? Please add this information to the table legend.
Answer: We thank the Reviewer for raising this point. The total number of samples submitted to RNA extraction, shown in Tables 3 and 4, derives from samples collected at multiple time points of sample collection from all included patients during the period of the study. We have added this information to the tables, as suggested by the reviewer.
- 4.Did you detect any mutations in the immunosuppressed patients that showed prolonged viral shedding?
Answer- We thank the Reviewer for this observation. Analysis of all viral genome sequences obtained from clinical specimens from all included patients did not reveal any mutations which could be associated with prolonged viral shedding, viral replication, or pathogenicity. We have included this observation in the text. Please check Results (lines 330-332).

Reviewer 3 Report
This is a very nice observational study that recounts some interesting data on a two groups of patients with COVID-19. However, I feel that there a number of places where this could be improved to make best use of the data (it appears that most of this may be in their data files):
1. I feel there is a bit of "over analysis" in places because there are lots of statistics on seemingly benign data. For example - is it of any real significance that one group is significantly older than the other? I would remove the stats and simply present the numbers, unless they are being used to make specific points.
2. For the virus culture experiments - there is no mention of any filtering or bacterial contamination. Were any of the failures to culture the virus due to bacterial contamination of the cultures? Although I won't go as far as to say that these data are not of value - I am not totally convinced they are needed if there are methodological problems and the few that grew out were 'lucky' samples with low contamination.
3. Instead of positive/negative: do the authors have any sense of viral load (by RNA copies per volume)? I believe they used a standard curve that should give these numbers.
4. Relatedly - table 8 would, I believe, be presented as a graph with time and viral load (or s% positive, or something). As a table -this is quite hard to follow and doesn't highlight the drops or changes.
I have put "major revision" because I believe this will take some rewriting and reanalysis - but, if the data is already available, this may not be a major issue.
Author Response
Answers to Reviewer 3
- I feel there is a bit of "over analysis" in places because there are lots of statistics on seemingly benign data. For example - is it of any real significance that one group is significantly older than the other? I would remove the stats and simply present the numbers, unless they are being used to make specific points.
Answer: We appreciate the reviewer ‘s comments and observations. Nevertheless, we believe that a detailed description of the study population may help the reader to understand the characteristics of the included patients. As mentioned in the Introduction section of the manuscript, different studies on SARS CoV-2 infection have suggested that the magnitude and duration of viral shedding correlates with disease severity, patient age and sex, viral load, etc (1-5). Taking into consideration that we intended to compare two groups of patients regarding duration of viral shedding and replication, we thought it would be important to acknowledge differences in specific characteristics among the two groups involved. For these reasons we think it would be important to maintain the statistics as presented in the original version of the manuscript.
- For the virus culture experiments - there is no mention of any filtering or bacterial contamination. Were any of the failures to culture the virus due to bacterial contamination of the cultures? Although I won't go as far as to say that these data are not of value - I am not totally convinced they are needed if there are methodological problems and the few that grew out were 'lucky' samples with low contamination.
Answer-We thank the Reviewer for this observation and for the opportunity to detail a little bit more the methodology involved in viral culture at our laboratory.
As detailed in the Methods section, at our laboratory viral culture for SARS-CoV-2 is conducted in a biosafety level-3 facility, where we utilize Vero CCL81 cells (ATCC® CCL-81™) in Dulbecco minimal essential medium supplemented with 5% heat-inactivated fetal bovine serum and antibiotics/antimycotics (lines 129-133).
This type of supplementation aims to prevent cell culture contamination, which is not a common event at our facilities.
Among 120 samples submitted to virus culture (Table 5) in the present study, we observed contamination in 8 samples (one anal swab and seven nasal swabs). When this type of event occurs, we follow a specific protocol, as previously described1-3. Briefly the contaminated isolates were submitted to two treatments; initially, microfiltration using a 0.22µ millipore filter (Minisart® 17761-ACK – Sartorius Stedim) which eliminates most bacteria (1). After this initial step, to the filtered material, 1% penicillin-streptomycin is added to eliminate bacterial cells (2-3).
Among 8 cases, after the use of this specific protocol, viral isolation was finally observed in two samples (nasal swab samples).
In summary, taking into consideration our previous experience with viral cultures and the protocol we follow at our institution we do not believe that contamination of viral cultures was associated with failure to observe viral isolation from clinical samples included in our study.
We did not include this type of information in the manuscript since this type of laboratory procedure is a routine procedure in virology laboratories.
We will be happy to include complete information regarding this aspect of the study in case the Reviewer considers it necessary.
1-Hahn MW. Broad diversity of viable bacteria in 'sterile' (0.2 microm) filtered water. Res Microbiol. 2004 Oct;155(8):688-91. doi: 10.1016/j.resmic.2004.05.003. PMID: 15380558.
2 - Araujo DB, Machado RRG, Amgarten DE, Malta FM, de Araujo GG, Monteiro CO et al. SARS-CoV-2 isolation from the first reported patients in Brazil and establishment of a coordinated task network. Mem Inst Oswaldo Cruz. 2020 Oct 23;115:e200342. doi: 10.1590/0074-02760200342.
.3 - Ge XY, Li JL, Yang XL, Chmura AA, Zhu G, Epstein JH et al. Isolation and characterization of a bat SARS-like coronavirus that uses the ACE2 receptor. Nature. 2013 Nov 28;503(7477):535-8. doi: 10.1038/nature12711.
- Instead of positive/negative: do the authors have any sense of viral load (by RNA copies per volume)? I believe they used a standard curve that should give these numbers.
Answer- We thank the Reviewer for this observation.
Briefly, the objective of the study was to investigate the presence and potential infectivity of SARS-CoV-2 in different biological specimens among patients with COVID-19.
The presence (persistence) of SARS-CoV-2 was defined based on the time spam (days) from the onset of symptoms to the last positive results that samples remained RT-PCR positive (Methods, lines 162-165).
Therefore, taking into consideration the objectives and methods proposed, we used positive/negative RT-PCR in Results and Discussion.
However, as mentioned in Methods (lines 110-126) quantitative assays (SARS-CoV-2 N or E gene) for SARS-CoV-2 were performed according to protocols adapted with primers and probes for the RT PCR assay [17-19].
Sequences of oligonucleotides were resuspended in known concentrations (serial dilution to base 10) for use as a positive control and for the construction of viral load quantification curves. The synthetic oligo sequences designed for the current study were based on a protocol previously described for other pathogens [21]. The RT PCR data was expressed as the value of the Cycle threshold (Ct), corresponding to the initial amplification cycle, which is inversely proportional to the number of copies of the target sequence of interest, given by the measurement of the number of copies per reaction.
- Relatedly - table 8 would, I believe, be presented as a graph with time and viral load (or s% positive, or something). As a table -this is quite hard to follow and doesn't highlight the drops or changes.
Answer- We thank the reviewer for his assessment. However, we respectfully disagree and believe that the data are most clearly expressed in a Table format. The other two reviewers apparently had no problem with this presentation.

Round 2
Reviewer 3 Report
I think it would be good to include more detail on the loss rate due to contamination and also the detailed method for how this was controlled. I fully accept that they had a very low rate (due to the work they did) - I just feel it is important to acknowledge that so that everything is clear.
However, I am happy to concede the other points. They are mainly presentational.
Author Response
I think it would be good to include more detail on the loss rate due to contamination and also the detailed method for how this was controlled. I fully accept that they had a very low rate (due to the work they did) - I just feel it is important to acknowledge that so that everything is clear.
Answer-We thank the Reviewer for this observation and for the opportunity to improve the quality of the manuscript.
We have added the information as suggested.
Please check the Results section which has been rewritten (lines 280-287).
